# Circadian Variation of Root Water Status in Three Herbaceous Species Assessed by Portable NMR

**DOI:** 10.3390/plants10040782

**Published:** 2021-04-16

**Authors:** Magali Nuixe, Amidou Sissou Traoré, Shannan Blystone, Jean-Marie Bonny, Robert Falcimagne, Guilhem Pagès, Catherine Picon-Cochard

**Affiliations:** 1INRAE, UR QuaPA, F-63122 Saint-Genès Champanelle, France; magali.nuixe@inrae.fr (M.N.); shannan.blystone@inrae.fr (S.B.); jean-marie.bonny@inrae.fr (J.-M.B.); guilhem.pages@inrae.fr (G.P.); 2INRAE, ISC AgroResonance, F-63122 Saint-Genès-Champanelle, France; 3Université Clermont Auvergne, INRAE, VetAgro Sup, UREP, F-63000 Clermont-Ferrand, France; robert.falcimagne@inrae.fr

**Keywords:** *Dactylis glomerata*, leaf water potential, low-field NMR, *Medicago sativa*, *Plantago lanceolata*, rhizotron, soil humidity, time domain NMR

## Abstract

Roots are at the core of plant water dynamics. Nonetheless, root morphology and functioning are not easily assessable without destructive approaches. Nuclear Magnetic Resonance (NMR), and particularly low-field NMR (LF-NMR), is an interesting noninvasive method to study water in plants, as measurements can be performed outdoors and independent of sample size. However, as far as we know, there are no reported studies dealing with the water dynamics in plant roots using LF-NMR. Thus, the aim of this study is to assess the feasibility of using LF-NMR to characterize root water status and water dynamics non-invasively. To achieve this goal, a proof-of-concept study was designed using well-controlled environmental conditions. NMR and ecophysiological measurements were performed continuously over one week on three herbaceous species grown in rhizotrons. The NMR parameters measured were either the total signal or the transverse relaxation time *T*_2_. We observed circadian variations of the total NMR signal in roots and in soil and of the root slow relaxing *T*_2_ value. These results were consistent with ecophysiological measurements, especially with the variation of fluxes between daytime and nighttime. This study assessed the feasibility of using LF-NMR to evaluate root water status in herbaceous species.

## 1. Introduction

Grasslands sequester high amounts of carbon in their soils [1,2,3], enabling them to potentially mitigate the concentration of greenhouse gases in the atmosphere [4]. The first step of this process is carbon fixation by photosynthesis in plant leaves, which is tightly coupled, at the leaf level, with transpiration, i.e., the outgoing flux of water in a plant. Thus, carbon sequestration processes depend on plant’s water fluxes in the soil–plant–atmosphere continuum.

On the scale of individual plants, water is moved through the soil–plant–atmosphere continuum via a variation in water potentials, the cohesion of water molecules and the regulation of stomatal aperture. Thus, under optimal edaphic water conditions, water flows from areas of higher water potential (soil) to areas of lower water potential (air) [5]. This differential of water potential pulls water from the soil into plant roots, up through the vascular system, and out of stomata in the leaves, thereby impacting the entire water status of the plant. Plant water fluxes also vary according to external parameters like radiation, soil water availability or plant characteristics such as total leaf area, root density, root conductance, root phenology, as well as circadian rhythm [5].

Roots are at the core of plant water dynamics and water status by enabling water uptake from the soil. During the day, water flows through the root cortex, via both the symplastic and apoplastic pathways, and then into the xylem upon entering the stele. During the night, the water flux is lower than during the day [6]. Nonetheless, root morphology and functioning are not easily assessable without destructive approaches, e.g., excavation and washing. The estimation of their water dynamics often requires indirect measurements on soil or leaves. Currently, only a few non-invasive methods exist to visualize root architecture and to study water uptake, such as 2D light transmission imaging combined with modelling [7,8], neutron radiography [9,10], X-ray tomography [11] and high field magnetic resonance imaging (HF-MRI) [11,12,13,14,15,16,17]. However, such measurements are performed on very simplified models, e.g., gel or sand to mimic the soil, and on a limited number of species.

Nuclear Magnetic Resonance (NMR), and its imaging counterpart (MRI), are interesting noninvasive approaches to study water status in plants. Indeed, by studying the most abundant molecule in living systems, i.e., water, NMR sensitivity issues can be overcome. ^1^H NMR is a quantitative method allowing the determination of water content, signal amplitude being directly related to the amount of water protons. By exploiting the high dependence of NMR parameters (relaxation times *T*_l_ and *T*_2_, diffusion coefficients) on how the water molecule is translationally hindered and rotationally bound, NMR stands as a valuable method to probe the multiscale status of water and its distribution in plants. For example, the water associated with tissues shows a low mobility and so a short relaxation time *T*_2_, whereas water which can be transported has a longer *T*_2_ [18,19]. Furthermore, thanks to the noninvasiveness of the method, follow-up studies aimed at characterizing changes in tissue structure and local water distribution can be conducted, either in response to water deprivation [20,21] or during the transformation of plant-based foods [22].

Unfortunately, it is impossible to perform acquisition in situ, i.e., directly in the plant’s natural environment with standard MRI systems, as they are not mobile. To displace the magnet, a low field MRI (LF-MRI) with a static magnetic field (**B**_0_) lower than 1 Tesla (T) must be considered. LF-MRI has been applied to plants to perform several analyses in the laboratory such as the visualization of organ structure [23,24] or phloem and xylem flux measurements [6,25,26,27,28]. For example, in 2006, Windt et al. [6] used a 0.72 T NMR spectrometer to measure xylem and phloem fluxes in four different species: poplar, castor bean, tomato and tobacco. They observed differences concerning the diurnal cycle, fluxes being slower at night than during the day and phloem flux being slower than the xylem flux. They also showed that the linear velocity and volumetric flow, especially of the xylem, vary greatly between species. LF-MRI permits the study of plant dysfunctions, such as vascular embolism, and their impact on the fluxes and the distribution of water in plants [24,29]. In addition to these indoor studies, others have been performed outdoors, in the plant’s natural environment [29,30,31]. However, with these previous magnets, the sample diameter was limited by the magnet bore. Unilateral magnets like the Nuclear Magnetic Resonance Mobile Universal Surface Explorer (NMR-MOUSE^®^) permit the study of a sample regardless of its size [32]. Even if the measurement depth is limited because of the sensitivity decrease with increasing distance from the surface of the magnet, plant organs other than stems or seeds can be explored using these instruments, e.g., leaves [21]. Despite this, roots are still under-studied. Recently, Bagnall et al. used LF-MRI to image roots in different types of soil, enabling the visualization of root architecture and morphology in the field and the characterization of the NMR properties of soil and roots [33]. They reported a difference in relaxation times between root and soil water, but also between the different types of soil. In addition, they observed an increase of soil water relaxation times with an increase in soil water content, regardless of the soil used. As plant roots are localized in soil, a challenge is the differentiation of water in both compartments. Thanks to the difference in relaxation times between soil and root water [15,33], the differentiation of these two compartments can be achieved. Here, we overcame this problem by using a model consisting of a rhizotron with a soft, thin textile in order to physically separate roots from soil while still enabling water and nutrient exchanges (for more details about the experimental design, see the description in Material and Methods).

To the best of our knowledge, there is no publication addressing the circadian variations of water status in plant roots using LF-MRI. Therefore, the aim of our study was to assess the feasibility of using a portable NMR spectrometer to characterize water status in roots by studying three herbaceous species grown in rhizotrons, having contrasting structure–function relationships. We seek to validate NMR measurements by comparing them to root morphologies and ecophysiological methods. In addition, we investigated the transverse relaxation time evolutions to have a better understanding of the dynamic of the water in roots.

## 2. Results

### 2.1. Climatic Chamber

Figure 1 shows the temporal evolution of the climatic chamber environmental parameters. A 14 h light cycle was applied with lamps on from 8:00 a.m. to 10:00 p.m., and an intensity of photosynthetic active radiation (PAR) of 560 µmol m^2^ s^−1^, and off during 10 h. Relative air humidity ranged from 48–53% during the day to 75–78% at night, whereas air temperature was, on average, 21 °C during the day and 18 °C at night for the three species. Air CO_2_ concentration was more variable due to entry into the chamber to perform ecophysiological measurements. It varied, on average, between 410 µmol mol^−1^ during the day and 550 µmol mol^−1^ at night.

### 2.2. NMR Profiles within the Three Rhizotrons

Figure 2 shows day-versus-night signal intensity profiles, i.e., the NMR signal intensities (the mean of all 256 echoes of the decay curves) as a function of the measurement depths, for the three species one day after watering. In Figure 2, the profiles clearly displayed a “1D image” of water in each rhizotron. Indeed, proton status in roots and soil yielded an imaging contrast, allowing the unambiguous attribution of water signal coming from each system. For clarity purposes, we only kept five points in the transparent wall (with signal void). For all species, the highest water signal intensity was observed in the roots (see Material and Methods). Thus, knowing the rhizotron organization, the remaining zone on the right of the roots with a signal slightly higher than zero corresponded to the soil. It is also noteworthy that each species displayed a distinct characteristic feature in their water signal profile. Firstly, a bimodal shape was clearly observed for the three species, with a maximum signal intensity of 0.09 and 0.18 for the left and right peaks of *D. glomerata,* of 0.09 and 0.11 for the left and right peaks of *P. lanceolata* and, of 0.27 and 0.18 for the left and right peaks of *M. sativa* (daytime). A sharp inflexion was only observed for *D. glomerata* and *M. sativa* at depths of 1.5 mm and 5.8 mm respectively. Secondly, the highest peak appeared to the right near the textile for *D. glomerata* and *P. lanceolata*, while the inverse was observed for *M. sativa*. Finally, *D. glomerata* and *P. lanceolata* displayed a thickness of 2.8 mm and 2.3 mm respectively, whereas the thickness of *M. sativa* was over 7 mm. The figure overlaid on each graph presents the root architecture within the measurement window (Figure 2). Figure 2 also shows that the profiles recorded during the day were slightly shifted in depth and had a lower amplitude than those acquired at night, regardless of the species.

### 2.3. Circadian Ecophysiological Measurements and NMR Signals in Roots and Soil

In order to observe the temporal evolution of the NMR water signal, a mean signal for both root and soil zones was calculated for each profile. The temporal evolutions of the leaf water potential (LWP), soil water content, and mean NMR water signal intensity in the roots and in the soil of the three species are presented in Figure 3.

The leaf water potential (Figure 3, top row) showed high variation between end-of-night (range of −0.18 MPa to −1.15 MPa) and end-of-day measurements (range of −1.16 MPa to −2.41 MPa), regardless of the species. At night, water potential increased (less negative values), corresponding to leaf rehydration when stomata were closed, and decreased (more negative values) during the day because of leaf transpiration. *D. glomerata* showed the highest fluctuations between day and night (mean calculated on day 2 and day 3: −1.98 MPa), whereas *P. lanceolata* had twice fewer fluctuations (~−0.86 MPa). In the case of *M. sativa*, on day 3, a moderate drought effect was observed as water potential measured at the end of the night reached −1.5 MPa. Close to irrigation events (black arrows), ecophysiological variables (leaf water potential, soil volumetric water content), as well as NMR signals in roots and soil, changed quickly, with pronounced peaks.

Over the full follow-up period, the root water signal intensity of the three species displayed similar patterns, with a decrease during the day and a slight increase at night (Figure 3, middle row). The decrease in the signal was more pronounced for *D. glomerata* the day following watering. Afterwards, a similar circadian variation, i.e., a decrease during the day followed by an increase during the night, was observed for the three species with an overall decrease in time.

The soil water signal intensity displayed globally similar circadian patterns as the root water signal intensity (Figure 3, bottom row) except that the water signal evolved in a plateau at night. Whatever the day and species, the NMR signal of the soil was always lower than that of the roots (10 times less). The figure shows the soil humidity whose evolution displayed a constant decrease over the whole follow-up period with a marked decrease during the day and a slow decrease at night.

### 2.4. Root Morphological Traits and Leaf Area

The three species showed contrasting morphologies. *M. sativa* differed from the two others with the highest root volume, root diameter and root mass, but the lowest root water content and intermediate total leaf area (Table 1). *D. glomerata* had the highest root length and leaf area, and the lowest root diameter. *P. lanceolata* had the lowest root length, root volume, root mass, and leaf area, but showed the highest root water content.

### 2.5. T_2_ Results

To complement circadian root water signal analysis, we performed *T*_2_ measurements by positioning the sensor at the depth displaying the maximum signal intensity in the profile of each rhizotron. Data analysis yielded two components. Figure 4 shows the time evolution of the two *T*_2_ values and their amplitudes over a 3-day-2-night follow-up period for the three species. For *D. glomerata* and *P. lanceolata* (Figure 4a,b), the fast-relaxing *T*_2_ component (short *T*_2_) relaxed with values around 5 ms, while the relaxation times for the slow relaxing component (long *T*_2_) varied between 60 and 80 ms. For these two species, the slow relaxing component represented the major fraction (from 70% to 95%). For these two species, only the long *T*_2_ value displayed a circadian rhythm with an increase during the night and a decrease during the day (from ~60 ms to ~80 ms). In contrast, the *T*_2_ analysis results for *M. sativa* showed different characteristics (Figure 4c). Firstly, the acquired echo decays were noisier than for the two other species. Consequently, only the first 30 CPMG signal decays (2 days/night periods) satisfied the signal-to-noise ratio (SNR) condition for NNLS analysis (see Appendix A) [34]. Secondly, while these inversions also resulted in two components, the slow relaxing one displayed lower *T*_2_ values, compared to the other two species, of around 50 ms. Thirdly, the two population fractions were closer together than those of the two other species. Finally, no visible circadian change was observed in any of the four parameters for *M. sativa*. During the T_2_ measurements, a high variation in LWP was still observed between the end of night (range of −0.5 MPa to 0 MPa) and the end of day for D. *glomerata* and P. *lanceolata* (−1.77 MPa to −0.53 (Figure 4a,b third column)) with similar amplitudes to those observed during the signal profile measurements (Figure 3, top row). For *M. sativa*, whilst the data were truncated at the beginning due to sensor problems, the LWP increased to −0.4 MPa at the end of the 2nd night, indicating that the plant was not in drought conditions.

## 3. Discussion

Our rhizotron model with a soft and thin textile enabled us to reduce the complexity of the study by separating roots from the soil, thus avoiding analysis of soil–root interactions, i.e., rhizosphere complexity [12,14,33], while getting closer to reality compared to other models such as hydroponics or agar plates cultures [35]. As the textile is very soft and weakly hydrophilic, the contacts between the roots, the textile and the soil are very high with no clear impact of the textile on plant development and morphology. Moreover, the use of a unilateral magnet, which is able to discriminate the different parts of our model, enabled us to be free of limitations imposed by model size and soil complexity.

The NMR-MOUSE is able to reveal a 1D image of water in the sample. Indeed, in agreement with the absence of any mobile protons in the transparent wall, the corresponding signal is void. Figure 2 also shows the depth profiles of water in the soil with a markedly lower signal intensity compared to the roots. The amount of water within the two compartments cannot fully explain this great difference in signal amplitude. Indeed, the roots of the three species were composed of an average of 75% water (Table 1), whereas the soil had a water content between 15–20% (Figure 3), yielding a ratio of approximately 4–5:1. This result was in line with previous studies obtained with high and low field NMR [14,33,36,37]. In Figure 2, we observed, on average, a ratio of 1:10 between the soil signal amplitude and the root signal amplitude. Therefore, we attributed this lower level of signal intensity in the soil as being a result of an interplay between the water binding capacity of the soil (which contains clay) and the effect of magnetic susceptibility inhomogeneity related to the heterogeneous structure of the soil. As each point of the profile corresponds to the mean of the 256 echoes of the decay curves, the profiles are also T_2_ weighted. The differences between root and soil NMR signals can be explained by the increased structure of water in a clay system versus a root system. On the other hand, water diffusion through the porous structure of the soil, combined with the strong magnetic field gradient of the NMR-MOUSE is known to have a deleterious effect on the acquired signal [14,15,33].

The photos in Figure 2 and the morphological data in Table 1 highlighted a great contrast in root architecture between the three species. Indeed, it appeared that *M. sativa* presented more coarse roots than the other plants as revealed by a higher mean root diameter (+75%) than *P. lanceolata* and *D. glomerata*. In addition, it had a total root mass that was 12.4 times and 3.2 times greater than *P. lanceolata* and *D. glomerata*, respectively, and a total root volume that was 7.2 times and 2.3 times greater than *P. lanceolata* and *D. glomerata*, respectively (Table 1). The higher proportion of coarse roots could explain why *M. sativa* had the lowest value of water content (12.6% and 16.5% lower than the mean water content of *D. glomerata* and *P. lanceolata*, respectively) as emphasized by some studies [38,39]. Therefore, the features of the depth profiles of the root water signal could be attributed to the root morphological traits of each plant. Indeed, the signal intensity at each depth is obtained by averaging the 256 echoes acquired with the CPMG pulse sequence to improve the SNR as commonly used in the inhomogeneous field [40]. These echoes are equivalent to the first 256 echoes of the CPMG acquisitions for T_2_ measurements. As will be discussed later, the amplitude of the signal at these early echoes is more weighted by water fractions in close interaction with root tissue, which is more abundant in *M. sativa*, in agreement with its high root mass and diameter. Furthermore, as watering was done in the soil compartment, and because roots use soil water, a water gradient from the edge of the soil to the transparent wall may be expected. Depending on the total root number present in the NMR window, roots localized close to the transparent wall were more disconnected from soil water than the ones in contact with the textile. Such a gradient, along with the proportion of small-sized (fine) roots, may potentially explain the bimodal shape of the *M. sativa* and *D. glomerata* root profiles. Considering potential differential root water contents of fine and coarse roots, we expect that the highest peak may be attributed to a higher proportion of fine roots while the inflection may originate from a higher proportion of coarse roots as observed in some grass species [38] or in diverse plant communities across different climatic zones [39].

The shifts in depth profiles between days and nights were related to an increase in the temperature of the magnet. Indeed, the well-known and strong temperature dependency of low field magnets results in the linear displacement of the measured volume. In our case, a shift of ~50 µm/°C was determined [41]. This agrees with the 200 µm shift observed in the present study as a consequence of the ~3 °C (18–21 °C) difference between the nighttime and the daytime. In addition, the reduction of the signal amplitude observed during the daytime is attributed, on the one hand, to the slight increase in the self-diffusion of water with an increase in temperature and, on the other hand, as will be discussed below, by an increase in water flux to satisfy plant transpiration.

Considering these day–night temperature effects, root and soil water signal variations can be monitored. With LF-NMR, we showed for the first time a circadian cycle measured in both roots and soil (Figure 3). Soil water NMR signal variation was consistent with soil humidity dynamics. The parallel evolution of the two parameters demonstrates that the NMR-MOUSE was able to monitor water status in the soil. Moreover, the circadian evolution of the root NMR signal was mirrored by the dynamics of leaf water potential and of soil–water content induced by plant transpiration. During the day, the upward transport of water, due to the transpiration flux, caused the decline in root NMR signal, whereas the root water uptake caused the decrease in soil NMR signal. Indeed, as leaves dehydrate during the day (decline of LWP), a similar diurnal pattern is expected for the roots, as shown by Huck et al. [42]; this is consistent with the presence of the soil–plant–atmosphere continuum. Moreover, nighttime evolution of all measurements corroborated this interpretation. Indeed, at night, water fluxes are slower than during the day [6] and the potential evapotranspiration is close to zero, explaining the increase of LWP near 0 MPa and the leaf and root rehydration. As a consequence, plants no longer absorb water, explaining the plateau of the soil NMR signal, soil humidity and the slight increase of root water NMR signal, which may be due to the decrease in proton mobility.

Our results also show that, besides these global traits shared by the three species, the NMR-MOUSE was also able to reveal specific structural and functional characteristics of each of the herbaceous plants. Indeed, as can be seen in Figure 3 on the middle row, the day after watering, root water signal amplitude declined by ~57% for *D. glomerata*, by ~9% for *P. lanceolata* and by ~18.5% for *M. sativa*, corresponding to a variation of LWP between the end of night and the end of day of −1.9 MPa, −0.8 MPa, and −1.1 MPa, respectively. These water signal changes were consistent with root morphological traits and with the total leaf area of each plant. *P. lanceolata*, having the lowest total leaf area, had a transpiration flux which was expected to be smaller than the flux of *D. glomerata*, which had a leaf area and root length almost three times greater.

In complement with the depth profile analysis, we performed transversal relaxation, *T*_2,_ analysis. *T*_2_ stands as the NMR parameter that most strongly indicates the contribution of different water fractions in the signal of complex systems like plants, and particularly the roots. By nature, the measured echo decays with the NMR-MOUSE are affected by the instrumental imperfections, as well as by the diffusion properties of the sample [43], resulting only in an effective relaxation time (*T*_2eff_) rather than a “true” *T*_2_. The distribution of *T*_2_ relaxation obtained with the regularized NNLS inversion [44,45,46] of the transversal decay signal of a complex system may be used to quantitatively analyze the state of water in each subsystem. Considering our model separating roots from soil, along with the root structure and function, the fast-relaxing T_2_ and its population fraction can be attributed to the water fraction in close interaction with root tissue, while the slow-relaxing T_2_ and its population fraction to the more mobile water fraction, i.e., water available to satisfy plant transpiration. Similar distribution was reported by Capitani et al. in their outdoor study using a unilateral NMR instrument to detect the water status in the leaves of different plants in stressed and unstressed conditions [21]. Because the *T*_2_ values of the slow relaxing water fraction of *D. glomerata* and *P. lanceolata* displayed a circadian feature and their population did not change, their variations could be ascribed to the daytime increase in water fluxes due to plant transpiration. Indeed, due to the strong magnet inhomogeneity, *T*_2_ measurements are highly sensitive to both coherent and incoherent water motions, and so the resulting incomplete refocalisation of spins (to the echo formation) leads to either signal and/or *T*_2_ reductions. The dependence of the CPMG echoes amplitude on diffusion and flow is well known since the discovery of the spin-echo for *T*_2_ measurement by Hahn in 1950 [47] and the paper by Carr and Purcell in 1954 [48]. Here, our analysis was limited with regard to the qualitative interpretations, the detailed analysis of the contribution of each phenomenon (i.e., flow and diffusion) being out of the scope of this paper and may be found in the above-cited articles as well as in others [49,50]. Conversely, the *T*_2_ value was higher at night as both temperature and plant transpiration decreased, whereas the absence of a clear variation observed for the short *T*_2_ agreed with the high interaction of this water fraction with root tissue. The effect of daytime flow was also clearly illustrated in the CPMG maximum signal amplitude of both *D. glomerata* and *P. lanceolata,* which clearly displayed a circadian variation (Appendix A). The fact that the main root water signal (up to 90%) of these two species is represented by this slow-relaxing water fraction (with a *T*_2_ value changing according to transpiration fluxes) along with the absence of any variation in the fast-relaxing (*T*_2_ and population) fraction stand as indicators of the well-watered status of these root tissues. This assumption is further reinforced by the results obtained on *M. sativa*. Indeed, either the SNR of the *T*_2_ signal decay (Appendix A) or the relatively high population of the fast-relaxing water fraction seems to indicate that this species was in a different hydric state to the two other species. The absence of any circadian change might be explained by perturbations in the transpiration function related to its water status.

To conclude, we assessed the feasibility of portable MRI to characterize the circadian dynamics of root water in three contrasted herbaceous species grown in rhizotrons, NMR measurements being validated with regards to the current ecophysiological reference methods. More studies are necessary to compare a wider range of species in order to define root water strategies in well-watered and droughted conditions. This study opens the opportunity to work with a rhizotron system in which roots are growing in the soil in order to better approximate the natural growing conditions.

## 4. Materials and Methods

### 4.1. Plant Material

#### 4.1.1. Rhizotrons

In November 2019, 3 flat parallelipedic containers called rhizotrons (95 × 40 × 5 cm), each with one transparent wall (in Plexiglass), were filled with a dried air granitic brown soil (12% clay, 17% silt, 59% sand, 13% organic matter), extracted from an upland grassland (St Genès Champanelle, 45.43° N, 03°10 E, 890 m a.s.l.) and sieved at 7 mm (pH ≅ 6.5). Before filling the rhizotrons, the soil was filled with slow-release fertilizer (35 kg m^−3^ NPK 14-7-14, Multicote 12, Haifa, Israel). Holes were also drilled at the bottom of each rhizotron and a pozzolan layer was added to allow for drainage.

A thin (60 µm) and soft tissue (Nylon Polyamide made, 100 × 45 cm) with a mesh of 30 µm and with a 20% open area of the pores was placed between the soil and the transparent wall in order to separate the roots from the soil, but allowing the transfer of water and nutrients. The transparent wall (4 mm thick) allowed closure of the rhizotrons with screws with the box and was then covered with black plastic between observations to shield the roots from light.

#### 4.1.2. Plant Material

One *Plantago lanceolata* plant and three tillers of *Dactylis glomerata* were transplanted from the site of St Genès Champanelle. Three *Medicago sativa* (Maga variety) plants were germinated from seeds. The rhizotrons were left outside during the winter and spring seasons before starting measurements in a climatic chamber during the summer season. A first cut of the plants at a height of 5 cm occurred on the 13 April in order to regenerate the leaves, as it occurs in mown grassland.

#### 4.1.3. Climatic Chamber

Climatic chamber environmental conditions were monitored and recorded at 30-s intervals with a data logger (CR6-Wifi, Campbell Scientific Ltd., Loughborough, UK) and averaged over 5-min periods for radiation and CO_2_ concentration (CARBOCAP, GMP343, Vaisala, Finland) and with a HOBO data logger (ONSET, Bourne, MA, USA) every ten minutes for relative air humidity and temperature. The chamber had a day and night cycle, with lights turning on at 8:00 a.m. and turning off at 10:00 p.m. Temperatures were maintained at around 21 °C during the day and 18 °C at night. Light values were measured with a PAR (JYP1000, SDEC, Reignac sur Indre, France).

### 4.2. Ecophysiological Measurements

#### 4.2.1. Leaf Water Potential

Psychrometers (PSY1-Stem, ICT International, Armidale, Australia) were used to measure leaf water potential (MPa). These measurements provided an insight into the daily oscillations of leaf water potentials under normal water conditions with regard to the magnitude of change between day and night measurements and with regard to inter-daily fluctuations. Measurements were made for each species on the following periods: 23 July–5 August 2020: *P. lanceolata*; 7–12 August 2020: *D. glomerata*; 31 August–8 September 2020: *M. sativa*. Measurements were made on two leaves for each species and the values were averaged.

#### 4.2.2. Soil Humidity

One 5-cm long sensor (EC-5, Meter Group, Pullman, WA, USA) was placed horizontally at one depth (16 cm) and connected to a datalogger (EM50, Meter Group, Pullman, WA, USA) to measure soil humidity continuously every 15 min, in all rhizotrons.

#### 4.2.3. Destructive Samplings

During the total sampling of the rhizotron after the NMR experiment, the total leaf area of green leaves was measured with an area meter (Licor 3100, Licor, Lincoln, NE, USA). After finishing all measurements on each species, the rhizotrons were harvested and plants were cut and sorted by organ type: leaves, stems, floral organs, and dead matter for each. In addition, roots were also sampled. Roots present inside the NMR measurement window (5 × 5 cm) were collected, washed and stored in a plastic bag (−18 °C) before performing root morphology measurements. For the evaluation of the root water content, fresh and dry roots were weighed separately. All fresh organs were oven-dried (48 h at 60 °C) and weighed to determine their dry mass (g). Mean root water content (RWC) was calculated as: $RWC=\frac{\left(fresh\;mass-dry\;mass\right)}{fresh\;mass}$ (g g^−1^).

#### 4.2.4. Root Morphology

To increase the contrast for scanning, the defrosted roots were stained with methylene blue dye (5 g L^−1^) by soaking them for at least one hour at ambient temperature. After rinsing in water to remove the excess stain, the roots were carefully separated into coarse (>1 mm) and fine roots and spread separately in a layer of water 1–3 mm deep, in a glass tray, using mounted needles. They were scanned with a flatbed scanner (EPSON perfection V700; Seiko Epson Corp., Suwa, Japan) at a resolution of 800 dots per inch, using the transparent mode. For each species, 6 to 10 images were recorded and thereafter analyzed with WinRhizoPRO software (V2012b, Régent Instruments, Québec, QC, Canada) to determine root length (m), average root diameter (mm), and root volume (cm^3^) by diameter class (10 classes of 0.1 mm-wide increments). The root volume was calculated as the sum of each volume by diameter class to avoid bias due to a skewed root diameter distribution [51].

### 4.3. NMR Experiments and Signal Analysis

#### 4.3.1. NMR-MOUSE System

NMR measurements were performed using a 0.3 T NMR-MOUSE (Magritek, Wellington, NZ) spectrometer. The full design of this low field system NMR sensor operating at a ^1^H resonance frequency of approximately 13.23 MHz, can be found elsewhere [32]. Briefly, this sensor is equipped with a permanent magnet whose configuration results in a strong gradient of approximately 3.2 T/m along the **B**_0_-(z)-direction, out from the surface of the magnet. Combined with a linear surface coil for radiofrequency (RF) transmission and signal reception, this linear gradient allows a selective signal measurement within a flat sensitive volume (on-resonance frequency) of a few tens of micrometers at a fixed distance of 25 mm from the magnet surface. In our configuration, a 4 × 4 cm RF coil (defining the spatial Field of View of the sensitive slice) was placed on top of a 10-mm thick spacer, which was positioned on top of the magnet, resulting in a distance of 15 mm between the coil and the sensitive volume. Measurements of different depths were possible via a high precision lift, which moved both the magnet and the RF coil downward along the z-direction from 0 to minus 15 mm, shifting therefore the measurement slice inside the sample from 15 mm to 0 (the sample/NMR-MOUSE interface). The NMR-MOUSE was fixed on a vector specifically designed for either transportation or to easily position it in contact with the sample thanks to the screws within the crank.

#### 4.3.2. Intensity Profile Measurements and Signal Analysis

For each rhizotron, the targeted measurement zone was selected visually through the transparent wall according to its root density. The NMR-MOUSE was then positioned in contact with the wall with its measurement window in front of the targeted measurement zone. The system was then securely clamped. The depth profiling protocol consisted of continuously acquiring the signal from the initial depth of 14.7 mm (the maximum measurement depth) to the final depth of 0 mm (the surface of the spectrometer) with a resolution of 0.1 mm by shifting the lift position in −0.1 mm-steps, resulting therefore in a total of 148 points for each profile. The signal of each depth consisted of the acquisition of 256 echoes using the CPMG pulse sequence with the following parameters: excitation pulse 15 μs, echo time 113 μs, repetition time 3000 ms. Each measurement was repeated 4 times for signal averaging at the exception of *P.lanceolata* for which 8 accumulations were performed, resulting in a duration of 37 min to record one profile for *D. glomerata* and *M. sativa* and of 1h10 for *P. lanceolata*. All acquisition parameters being identical, the NMR signal of *P. lanceolata* was divided by two to be compared to the others profiles. A gap of 1 min was set between two successive profiles to allow the lift to return to its initial position. To construct the signal profile, the signal intensity at each position was derived by the mean of the 256 echoes. An example of the result of the 3-day acquisition for the *D. glomerata* sample is presented in Figure 5. All of these profiles clearly display three spatially separated parts attributed to the soil, roots and the transparent wall according to the dimensional characteristics of the sample. Indeed, from 0 to 4 mm, the null signal intensity (noise) perfectly reflects the absence of any mobile protons in the 4 mm-transparent wall. The signal increase observed in the forward depths corresponded to the protons (mainly from water) in the roots. The signal decrease (the average of the 256 echoes) observed at 7.5 mm due to the textile followed by a low flat signal assigned to the water in the soil.

#### 4.3.3. T_2_ Measurements and Fitting

*T*_2_ measurements were performed at one unique position for each sample for the three species. After the profile measurements, the lift was shifted to the measurement volume corresponding to the position in the roots displaying the maximum signal intensity. The full transversal decay curve for that position was then obtained by recording 2800 echoes for *D. glomerata* and *M. sativa* and, 2500 echoes for *P. lanceolata* using the CPMG pulse sequence with the following parameters: excitation pulse 12 μs, echo time 100 μs and repetition time 12 s for *D. glomerata* and *M. sativa*, and 10 s for *P. lanceolata*. To improve the signal-to-noise ratio (SNR), each echo was averaged 128 times for *D. glomerata* and *P. lanceolata* and 256 times for *M. sativa*. Assuming that the measured echo decays consisted of a superposition of exponential decays, data analysis was performed in terms of a distribution of relaxation times. The *T*_2_ distribution functions were extracted from the experimental data using an in-house implementation in MATLAB^®^ (MathWorks, Natick, MA, USA) of the NNLS inversion algorithm using a regularization parameter to control the trade-off between stability and bias (Appendix A). When more than two peaks were present (that was the case for less than 5% of the data), and because we hypothesized a bi-exponential relaxation curve, a mean between values of the same order of magnitude was calculated.

## Figures and Tables

**Figure 1 plants-10-00782-f001:**
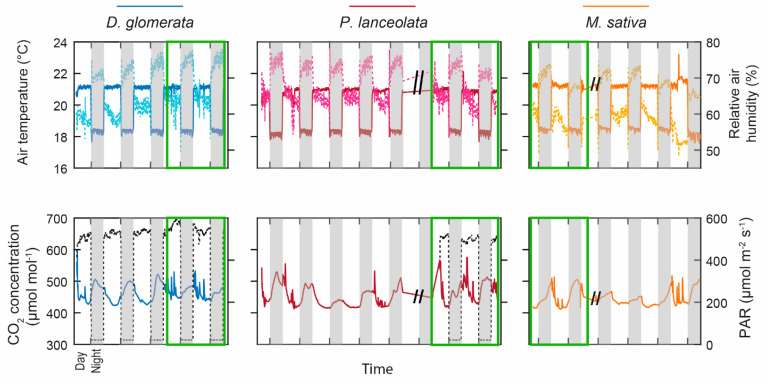
Climatic chamber parameters as a function of time during the experiment on the three species: *D. glomerata* in blue, *P. lanceolata* in red, and *M. sativa* in orange. The topmost plot represents the air temperature and the relative air humidity (solid and dotted light lines, respectively) and the bottom plot represents the CO_2_ concentration and the photosynthetic active radiation (PAR) in black dotted lines. The white and grey boxes represent the presence (8:00 a.m. to 10:00 p.m.) and the absence of light (10:01 p.m. to 7:59 a.m.), respectively. The green boxes correspond to the period of *T*_2_ measurements while the other data were obtained during the NMR profile measurements.

**Figure 2 plants-10-00782-f002:**
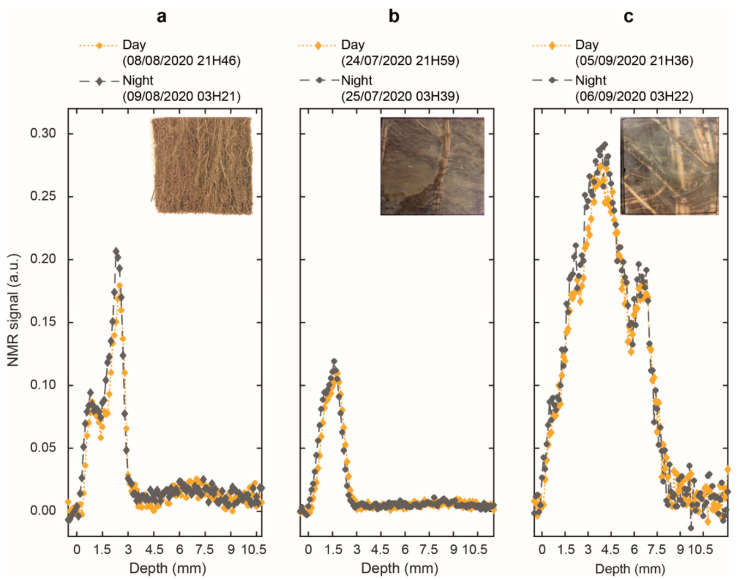
Day and night NMR signal intensity profiles (in yellow and grey, respectively) of (**a**) *D. glomerata*, (**b**) *P. lanceolata* and (**c**) *M. sativa* measured one day after watering. Overlaid on each graph, a picture showing the roots present within the NMR measurement window (outer section of 5 × 5 cm) with a sensitive (coil section) section of 4 × 4 cm.

**Figure 3 plants-10-00782-f003:**
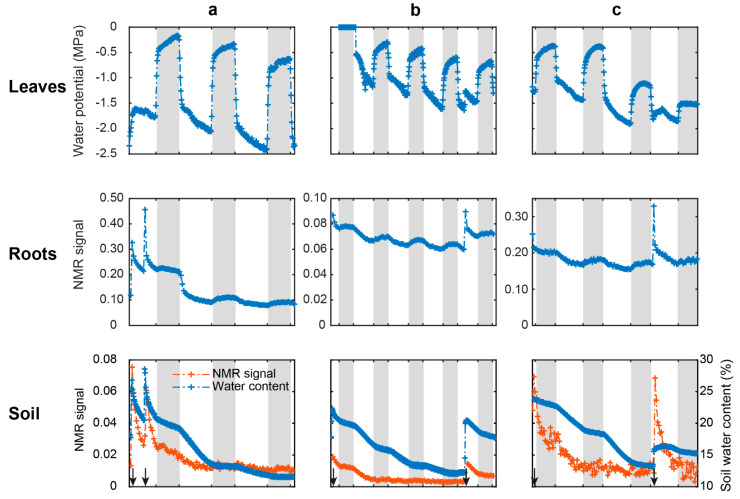
Evolution of the leaf water potential (top), the average NMR signal intensity measured in roots (middle) and in soil (bottom, orange) and soil volumetric water content (bottom, blue) measured in (**a**) *D. glomerata*, (**b**) *P. lanceolata*, and (**c**) *M. sativa*. The white and grey boxes represent the day and night periods, respectively and the black arrows at the bottom represent watering events.

**Figure 4 plants-10-00782-f004:**
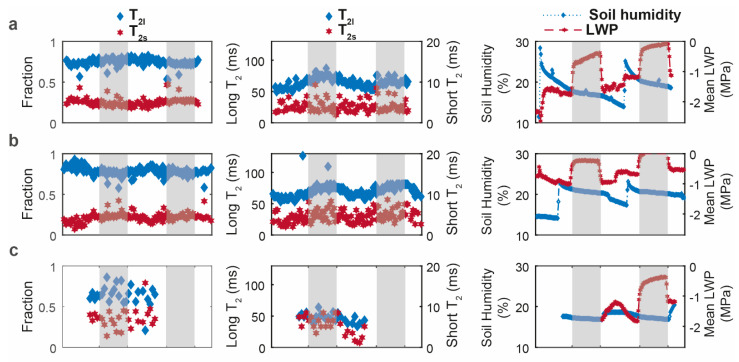
Evolution of the *T*_2_ proportions (first column) and of the *T*_2_ values (second column) in the root compartment, and of soil humidity and mean leaf water potential in blue and red, respectively (third column) for (**a**) *D. glomerata* (**b**), *P. lanceolata*, and (**c**) *M. sativa* (*T*_2s_: short *T*_2_; *T*_2l_: long *T*_2_). The white and grey boxes represent the presence and the absence of light, respectively.

**Figure 5 plants-10-00782-f005:**
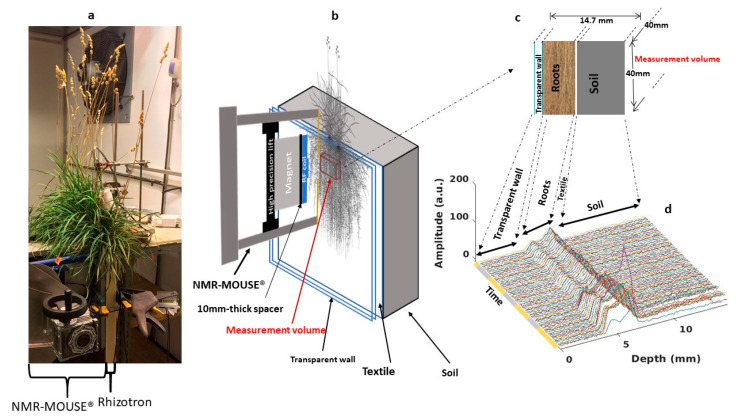
(**a**) Experimental setup in the climatic chamber (case of *D. glomerata*). (**b**) Schematic representation of the magnet in contact with a rhizotron and of the position of the NMR sensitive volume (or slice) (red rectangular parallelepiped). (**c**) Illustration drawing of different structures in the measurement window. (**d**) 3-day (daytime in yellow and nighttime in gray) profile, i.e., signal intensity (average of 256 echoes) at each depth, cycle. Soil, roots and transparent wall compartments are clearly revealed in each profile whereas no clear feature can be attributed to the textile as described in the text.

**Table 1 plants-10-00782-t001:** Root morphological traits extracted from the roots present inside the NMR measurement volume and the total leaf area (rhizotron scale) measured at the end of the NMR experiment during plant harvest for the three species.

Variables	*D. glomerata*	*P. lanceolata*	*M. sativa*
Total root length (m)	56.237	17.498	46.137
Total root volume (cm^3^)	2.431	0.780	5.647
Total root dry mass (g)	0.507	0.132	1.634
Mean root diameter (mm)	0.223	0.270	0.432
Mean root water content (g g^−1^) *	0.772	0.808	0.675
Total leaf area (cm^2^)	6055.8	2166.6	4976.9

* roots were washed before weighing the fresh root mass.

## Data Availability

The data presented in this study are openly available in Data INRAE (https://data.inrae.fr/, accessed on 12 March 2021) repository at https://doi.org/10.15454/NWRHDA (accessed on 12 March 2021).

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
