# Peer review of "Circadian Variation of Root Water Status in Three Herbaceous Species Assessed by Portable NMR"

_plants, 2021, doi:10.3390/plants10040782_

Round 1
Reviewer 1 Report
The manuscript presents the analysis of water status changes for three different plant root systems grown in rhizotrons using single sided NMR combined with conventional measurements. The research idea is novel, the setup reasonable, results and conclusions are presented in a comprehensive way. Concerning the analysis of the relaxation times, the authors limit themselves to a qualitative discussion which is absolutely convincing since the paper aspires to set the methodology for this type of investigation as basis for future work.
Alltogehter, is is worth to be published after some minor corrections. These are commented directly in the attached pdf document.

Author Response
Dear,
Please find in the attached file named "AuthorsResponsesToReviewer_1.pdf" our point-by-point responses to the comments by reviewer 3. We have choseen to respond to the reviewer comments directely in the same file for clarity. Changes are made accordigly in the manuscript tract. We ppreciated the kind suggestions by reviewer for improving the manuscript.
Best regards

Reviewer 2 Report
The manuscript describes the applications of the NMR Mobile Universal Surface Explorer system for the in situ studies of root water status in three species. The temporal evolutions of the total NMR signal and of the transverse relaxation time T2 were studied and the circadian variations of these parameters were correlated with the ecophysiological parameters. The results and discussion are consistent with the experiments that the authors have conducted. In my opinion, the study is of interest to the readership of Planta, and the work reported therein could potentially contribute to the development of protocols for in situ studies of root water status using the rhizotron and low-field portable NMR. So I think that the paper can be accepted in the present form.
Nevertheless, some small corrections could probably help the readers.
Results:
- Figure 1
- The “photosynthetic active radiation (PAR) in grey” (black dotted lines?) is not reported for M. sativa? So, was it the same as for the two other samples?
- It is not easy to distinguish the solid ant dotted lines of the same color (red/red, blue/blue…). Especially for P. lanceolate
- The Figure 1 presents the results in the following order: lanceolata, D. glomerata, M. sativa. In the figures 2, 3 and 4, the order is different. It is a little confusing for the reader.
- Figure 2. Title.
“Day and night profiles…”. To assist the reader, please specify in the title what kind of depth profile is represented here. (NMR signal intensity profiles?). What are the units of y-axis? Arbitrary units? Would it be interesting to report these values to the Fresh Weight of the roots contained in the “NMR measurement window”?
- Table 1. The mean root Fresh Weight is not given in this table neither in MM part. Could it be interesting to specify these values?
Discussion
Line 220. Please, specify the abbreviation LWP when used in the text for the first time.
Line 335. “The effect of daytime flow was also clearly illustrated in the CPMG maximum signal amplitude of both D. glomerata and P. lanceolata which clearly displayed a circadian variation (Figure S1).” To illustrate this statement it could be interesting to add the day/night boxes to the “Experiment number scale” of this Figure.
Materials and Methods
Figure 5. It is not easy to understand the experimental system with this Figure. The Figure 5 could be presented in Supp. Information and an enlarged picture with a better contrast and annotations (or a simplified sketch of the installation) could help the reader. Please, indicate in this Figure the z-axis direction cited in the text.
Figure 6. (b) : the blue rectangular parallelepiped indicates “the NMR measurement window position”. It could be interesting to specify here the size of this window (the readers can find this information in the Results (5x5 cm)). Otherwise, does the sensitivity of the probe vary across this 5x5cm region?
Author Response
Dear,
Please find in the attached file named AuthorsResponsesToReviewers_2. docx our reponses to the comments and suggested changes by the reviewer 2 on our manuscript. We appreciated the reviewer suggestions and have made our best to follow them.
Best regards

Reviewer 3 Report
This study explores using a small low-field NMR magnet (NMR Mouse) to monitor the circadian variation of the root water in three different vascular plants. The study exploits the versatility of a portable NMR magnet to analyze samples under their natural conditions; in this case, the growing roots under the soil. This is a highly cost-effective approach to acquiring informative (and in situ) results compared to a high-field MR study. For each plant, circadian results of the water content in roots and its T2 were acquired. Indeed, the methodology present here could open new opportunities to agriculture research.
Despite the solid methodology, the manuscript is a little hard to follow. I feel that the discussions could further strengthen.
-It is not clearly described in the overall setup of the experiment. The author should consider removing figure 5, and improve figure 6. For example, an illustration drawing showing the position of the magnet with respect to the rhizotron panel; and how the density profile (figure 2) corresponds to the samples inside the rhizotron (as shown in Figure 6d, but it is not clear and could be improved).
-L265-270: this discussion on the observed density profile in Figure 2 (and in L140-148) is not clear. What is the textile refer to? And the transparent wall? I suppose the so-called bimodal shape observed in Figure 2 is simply a coincidence. It depends on how the roots distribute (and the distribution density) across the rhizotron.
-L256: low water content in Sativa. This statement is not clear and confusing, because earlier, it claimed that the observed signal intensity in Figure 2 might be fully attributed to water (L132). Based on this, Figures 2 and 3 show contradiction to L256.
-T2 results and discussion: these results are encouraging. If I interpret the study correctly, the T2 measurements were carried out on a single slice with the highest water level in Figure 2 (i.e. with the highest root distribution?). It deduces two different T2 components (long and short) in Figure S2. The assignments (or speculations) of the two components are not clearly stated in the text. I ‘speculate’ that the long T2 ascribe to the water molecules inside (and outside of) the root tissue and that the short T2 correspond to the cellulose of the root tissue, or the bound water molecules with the cellulose? Are there literatures support the deduced T2 values? Can the observed distribution (line-width in Figure S2d) tell us about the root morphology, how do the line-widths (or line shape) compare in all three plants?
-could the observed T2 variations in Figure 4 also attribute to the change in temperature between the day and night, as discussed earlier in the text.
-the CPMG decay for Sativa (in Figure S1) looks strange. First, I suppose the experiment number correspond to the circadian measurement (from 1-50 or 1-128). For Sativa, it is interesting to find a drastic decrease in SNR (even on the 1st echo of the decay). Could this cause by a sudden change in the setup (i.e. magnet, rf tunning), or simply the H2O disappear?
Author Response
Dear,
Please find in the attached file named "AuthorsResponsesToReviewer_3.docx" our point-by-point responses to the comments by reviewer 3. We kindly appreciate the reviewer suggestion for improving the manuscript.
Best regards
